# Effective Functional Immunogenicity of a DNA Vaccine Combination Delivered via In Vivo Electroporation Targeting Malaria Infection and Transmission

**DOI:** 10.3390/vaccines10071134

**Published:** 2022-07-16

**Authors:** Yi Cao, Clifford T. H. Hayashi, Fidel Zavala, Abhai K. Tripathi, Hayk Simonyan, Colin N. Young, Leor C. Clark, Yukari Usuda, Jacob M. Van Parys, Nirbhay Kumar

**Affiliations:** 1Department of Global Health, Milken Institute School of Public Health, George Washington University, Washington, DC 20052, USA; yicao@gwu.edu (Y.C.); chayashi@gwu.edu (C.T.H.H.); leorcc@gwu.edu (L.C.C.); yukari.usuda@riken.jp (Y.U.); jmvanparys@gwmail.gwu.edu (J.M.V.P.); 2Department of Molecular Microbiology & Immunology, Johns Hopkins Malaria Research Institute, Bloomberg School of Public Health, Johns Hopkins University, Baltimore, MD 21205, USA; fzavala1@jhu.edu (F.Z.); atripat2@jhu.edu (A.K.T.); 3Department of Pharmacology and Physiology, School of Medicine and Health Sciences, George Washington University, Washington, DC 20052, USA; hayksimonyan@gwu.edu (H.S.); colinyoung@gwu.edu (C.N.Y.)

**Keywords:** *Plasmodium falciparum*, transmission blocking vaccine, pre-erythrocytic vaccine, combined DNA vaccine

## Abstract

*Plasmodium falciparum* circumsporozoite protein (PfCSP) and Pfs25 are leading candidates for the development of pre-erythrocytic and transmission-blocking vaccines (TBV), respectively. Although considerable progress has been made in developing PfCSP- and Pfs25-based vaccines, neither have elicited complete protection or transmission blocking in clinical trials. The combination of antigens targeting various life stages is an alternative strategy to develop a more efficacious malaria vaccine. In this study, female and male mice were immunized with DNA plasmids encoding PfCSP and Pfs25, administered alone or in combination via intramuscular in vivo electroporation (EP). Antigen-specific antibodies were analyzed for antibody titers, avidity and isotype by ELISA. Immune protection against sporozoite challenge, using transgenic *P. berghei* expressing PfCSP and a GFP-luciferase fusion protein (PbPfCSP-GFP/Luc), was assessed by in vivo bioluminescence imaging and blood-stage parasite growth. Transmission reducing activity (TRA) was evaluated in standard membrane feeding assays (SMFA). High levels of PfCSP- and Pfs25-specific antibodies were induced in mice immunized with either DNA vaccine alone or in combination. No difference in antibody titer and avidity was observed for both PfCSP and Pfs25 between the single DNA and combined DNA immunization groups. When challenged by PbPfCSP-GFP/Luc sporozoites, mice immunized with PfCSP alone or combined with Pfs25 revealed significantly reduced liver-stage parasite loads as compared to mice immunized with Pfs25, used as a control. Furthermore, parasite liver loads were negatively correlated with PfCSP-specific antibody levels. When evaluating TRA, we found that immunization with Pfs25 alone or in combination with PfCSP elicited comparable significant transmission reduction. Our studies reveal that the combination of PfCSP and Pfs25 DNAs into a vaccine delivered by in vivo EP in mice does not compromise immunogenicity, infection protection and transmission reduction when compared to each DNA vaccine individually, and provide support for further evaluation of this DNA combination vaccine approach in larger animals and clinical trials.

## 1. Introduction

It was estimated that in 2020, there were 241 million malaria cases and 627,000 malaria deaths worldwide, most of whom were children under 5 years of age [1]. The development of highly effective and durable vaccines against human malaria parasites remains a key priority. The malaria vaccine roadmap led by the World Health Organization (WHO) encompasses two strategic objectives by 2030: (i) developing vaccines with high protective efficacy (at least 75%) against clinical malaria, and (ii) developing vaccines that reduce transmission in order to substantially reduce the infection incidence [2]. Recently, WHO approved the first-generation malaria vaccine (RTS,S/AS01) for the prevention of *Plasmodium falciparum* in children living in regions with moderate to high transmission [3]. RTS,S/AS01 is a pre-erythrocytic vaccine (PEV) formulated using the adjuvant AS01, which targets the *P. falciparum* circumsporozoite protein (PfCSP) present on the surface of sporozoites inoculated by blood feeding mosquitoes [4,5]. It has been shown to induce both humoral and cellular immune responses [4,6], conferring reduction of clinical malaria and severe malaria in children. However, the immune protection has been shown to be partial, 46% against clinical malaria and 34% against severe malaria, and limited longevity in a phase III clinical trial [7]. More efficacious next-generation vaccines are still urgently needed to fill the gap in the strategic goals for 2030 set by the WHO malaria vaccine roadmap.

One widely considered strategy to develop more efficacious malaria vaccines is to include multiple antigens from different lifecycle stages, especially those which reduce both transmission and disease. Transmission-blocking vaccines (TBV) target antigens present in the mosquito/sexual-stages of the parasite and interfere with the process of sexual development in the mosquito vector, thus leading to transmission reduction [8,9]. TBVs are considered as not only an essential tool for malaria elimination but also an ideal way to increase vaccine efficacy when combined with pre-erythrocytic vaccines (PEV) or blood-stage vaccines. Several TBV antigens have been identified for vaccine development, including Pfs25, Pfs230 and Pfs48/45 in *P. falciparum* with homologues in *P. vivax* [10,11]. Pfs25 is the most advanced TBV candidate, which has been extensively evaluated in pre-clinical studies in various animals. However, several early clinical trials in humans failed to induce robust and long-lived antibody responses, in part due to the poor immunogenicity of monomeric Pfs25 protein. Several modifications of Pfs25-based vaccines have been made to enhance specific antibody response against Pfs25, such as chemical conjugation to itself or to the exo-protein A of *Pseudomonas aeruginosa* [12,13], fusion to self-assembly nanoparticle platform IMX313 [14], expression in virus-like particles or in viral-vectored platforms [15,16,17], and formulation in new adjuvants [18]. 

Both PEV and TBV are categorized as vaccines that have the potential to interrupt malaria transmission to support malaria elimination [8]. A recent study found that two passive transferred monoclonal antibodies, targeting pre-erythrocytic CSP and sexual-stage P25 at partially efficacious concentrations, functionally synergized to eliminate malaria parasites over multiple generations of mouse–mosquito transmission [19]. It further suggested the possibility that a combination of PEV and TBV, such as PfCSP and Pfs25, can enhance intervention efficacy to effectively eliminate the malaria parasite. Furthermore, addition of a PEV to a TBV will enhance acceptance of TBVs, which on their own lack a direct protective effect against malaria infection and disease while providing indirect protection via inhibiting overall malaria transmission at the population level [11]. In order to address this perceived need, studies have begun to evaluate the combination of vaccines targeting infection and transmission. A combination of RTS,S and Pfs25-IMX313, formulated using AS01 adjuvant, was shown to maintain the immunogenicity of both vaccines and mediate biological functions against parasites in both lifecycle stages when compared to Pfs25-IMX313 or RTS,S alone [20]. Using a viral-vectored platform, PfCSP and Pfs25 were expressed as single vaccines or as a fusion vaccine in human adenovirus 5 (AdHu5) and adeno-associated virus serotype 1 (AAV1). Heterologous AdHu5-prime/AAV1-boost immunization with a mixture of single-antigen vaccines or the Pfs25-PfCSP fusion vaccine elicited robust and durable antibodies against both PfCSP and Pfs25, which were effective for both protection and transmission-blocking activity [16,21].

The DNA-based vaccine approach represents one of the promising platforms for the development of malaria vaccines from bench-scale to clinic-scale use, due to ease of production, low cost, stability, and the ability to induce cellular and humoral responses [22,23]. Our previous studies have demonstrated the immunogenicity and functional potency of the Pfs25 DNA vaccine in mice and nonhuman primates [24,25]. We have shown improved functional immunogenicity of the Pfs25 DNA vaccine through codon-optimization and in vivo electroporation (EP) [26,27] and by a heterologous DNA prime-protein boost regimen in higher mammals [25]. Pfs25 DNA vaccine has also been evaluated by immunization in mice and nonhuman primates when combined with another TBV DNA vaccine, Pfs48/45, and specific antibody responses to both antigens and transmission-blocking activity were not compromised [28,29]. These studies provide compelling rationale for combining Pfs25 with other vaccine candidates targeting different lifecycle stages for developing more effective vaccines. DNA vaccines encoding PfCSP have been previously demonstrated to induce specific T cell responses and anti-CSP antibodies that can provide substantial protection from malaria infection in mice, monkeys and humans [30,31,32]. Many improvements in the PfCSP DNA vaccine in the last two decades have dramatically enhanced its immunogenicity and protective efficacy, including DNA priming and heterologous boosting with virus-vector or recombinant proteins [33,34], DNA vaccines containing a larger number of epitopes from CSP and other putative PEV target antigens [35], co-expression of plasmid-encoded cytokines [23] or molecular adjuvants [36], and co-administration of DNA vaccines encoding other malaria antigens.

The goal of this study was to evaluate the feasibility of a combination of pre-erythrocytic PfCSP and sexual-stage Pfs25 DNA vaccines delivered by in vivo EP in mice, compare the immunogenicity of combined DNA vaccines against individual vaccines, and assess the protective effects against sporozoite challenge and mosquito-stage parasite development. The successful outcome of an effective combination vaccine will provide a strong rationale for evaluation of similar combinations in larger animals and in clinical trials. 

## 2. Materials and Methods

### 2.1. DNA Vaccine Plasmids

Pfs25 sequence (AF193769.1) lacking N-terminal signal and C-terminal anchor sequences was codon-optimized for expression in mammalian cells and cloned into the DNA vector VR1020 (Vical, San Diego, CA, USA) downstream of tissue plasminogen activator signal sequence. All three putative N-linked glycosylation sites were not mutated in the Pfs25 coding sequence employed [24,25,26,27]. Full-length PfCSP sequence (XP_00135122.1) was codon-optimized for expression in mammalian cells and cloned into DNA vector VR1020 (GenScript, Piscataway, NJ, USA) [30,32,37]. Pfs25-VR1020 was produced and supplied in deionized water at 2.5 mg/mL and <30 EU endotoxin per mg DNA (Aldevron, Fargo, ND, USA). PfCSP-VR1020 was produced and supplied in deionized water at 2.5 mg/mL and < 0.1 EU endotoxin per µg DNA (Alta Biotech, Aurora, CO, USA). For immunization of the DNA combination, two vaccine plasmids were concentrated, mixed together with equal amounts of each plasmid, and diluted to a final concentration of 5 mg/mL.

### 2.2. Mammalian Cell Transfection and Western Blotting

Pfs25-VR1020 and PfCSP-VR1020 DNA plasmids were used to transfect mammalian Expi293F cells (Life Technologies, Carlsbad, CA, USA). Suspension cells were maintained in the Expi293 expression medium and transfected with individual plasmids using ExpiFectamine 293 transfection kit (Life Technologies, Carlsbad, CA, USA) as per the product protocol. At 72 h post transfection, the cell culture was harvested and centrifuged to separate supernatants and cells. Supernatants were filtered through 0.22 μm filter (MilliporeSigma, Burlington, MA, USA) to remove cell debris and concentrated by the centrifugal filter (MilliporeSigma, Burlington, MA, USA). Cells were washed twice with phosphate-buffered saline (PBS, pH 7.4), and lysed in cell lysis buffer (RIPA buffer, Sigma-Aldrich, St. Louis, MO, USA). The protein expression in the supernatants and cells was evaluated by Western blotting under non-reducing and reducing conditions using Pfs25-specific mouse immune serum and PfCSP-specific monoclonal antibody (2A10) [38]. 

### 2.3. Immunization Dose and Schedule

Balb/c and C57BL/6 mice, 6–8 weeks old (Charles River Laboratories, Wilmington, MA, USA), were used for immunization. Female Balb/c mice (n = 10 or 5 per group) were randomly assigned to six groups (Figure 1). In groups 1, 2 and 3 (n = 10), female Balb/c mice received Pfs25-VR1020, PfCSP-VR1020 and a combination of Pfs25-VR1020 and PfCSP-VR1020, respectively, via in vivo EP. In groups 4, 5 and 6 (n = 5), female Balb/c mice received Pfs25-VR1020, PfCSP-VR1020 and the combination, respectively, but without EP. To address sex as a biological variable, male Balb/c mice were randomly assigned to three groups (groups 7, 8 and 9, n = 5), and immunized via in vivo EP with Pfs25-VR1020, PfCSP-VR1020 and the combination, respectively (Figure 1). In addition, female C57BL/6 mice (group 10, n = 5) received Pfs25-VR1020 via in vivo EP to evaluate the immunogenicity of Pfs25 DNA vaccine in a different mouse strain (Appendix A).

Mice were anesthetized by isoflurane inhalation for DNA injection and EP. Fifty micrograms of each DNA plasmid in 20 μL sterile deionized water were administered by intramuscular injection in the tibialis anterior muscle. Mice in the DNA combination groups received 50 μg of Pfs25-VR1020 and 50 μg of PfCSP-VR1020 in a total volume of 20 μL water for each dose. EP was performed using a needle electrode array (4 × 4 × 5 mm) and an Agile Pulse ID Generator (BTX Harvard Apparatus, Holliston, MA, USA). All Balb/c groups received 3 DNA immunizations at 4-week intervals. Blood samples were collected on day 0 (pre-immune bleed, PB), 14 days after the 2nd immunization (bleed 1, B1), 10 days after the 3rd immunization (bleed 2, B2), and 14 days after the sporozoite challenge (final bleed, FB) (Figure 1). The C57BL/6 group received 4 DNA immunizations at 4-week intervals. Blood samples were collected on day 0 (PB) and 21 days after the second (B1), third (B2) and fourth immunizations (B3). These C57BL/6 mice were used as a negative control for sporozoite challenge studies and bled 14 days later (FB, Appendix A). Sera were stored at −20 °C until further use.

### 2.4. Assessment of Antibody Titers, Isotype and Avidity by Enzyme-Linked Immunosorbent Assay (ELISA)

Recombinant Pfs25 and PfCSP were expressed and purified in *Escherichia coli.* Expression, purification and refolding of recombinant full-length Pfs25 has been described previously [39]. The PfCSP sequence lacking N-terminal signal and C-terminal GPI anchor residues was codon-harmonized [40], fused with a 6x histidine tag at the 5′ end and a six amino acid spacer (PGGSGSGT), and cloned into the expression vector pET(K-). BL21(DE3) *E. coli* cells transformed with PfCSP-pET(K-) plasmid were grown to an optical density (OD600) of ~1.00, followed by induction with 0.1 mM IPTG for 3 h at 30 °C. The cells were harvested and lysed by microfluidization in lysis buffer (50 mM NaH_2_PO_4_, 300 mM NaCl, 20 mM imidazole, pH 8). Following centrifugation (20,000× *g* for 30 min at 4 °C, Beckman Avanti J-E), the supernatant was incubated with Ni-NTA beads (Qiagen, Germany) overnight at 4 °C. The beads were washed with the buffer (50 mM NaH_2_PO_4_, 300 mM NaCl, 20 mM imidazole, 0.25% sarkosyl, pH 8), followed by another wash without sarkosyl. Recombinant PfCSP was eluted from the beads with 250 mM imidazole in 50 mM NaH_2_PO_4_, 300 mM NaCl, pH 8. The protein was buffer-exchanged with PBS (pH 7.4), concentrated using the Amicon Ultra centrifugal filters (MilliporeSigma, Burlington, MA, USA), and stored at −20 °C. The quality of recombinant Pfs25 and PfCSP was assessed by SDS-PAGE and Western blotting, and protein concentrations were measured by the bicinchoninic acid (BCA) method (Thermo Scientific, Waltham, MA, USA).

Pfs25- and PfCSP-specific antibody titers were analyzed in the immune sera by ELISA. Briefly, NUNC Maxisorp 96-well plates (Thermo Fisher, Waltham, MA, USA) were coated with recombinant PfCSP at 100 μL/well (50 ng/mL) in PBS. The Immulon 4HBX 96-well plates (Thermo Fisher, Waltham, MA, USA) were coated with recombinant Pfs25 at 100 μL/well (1 μg/mL) in carbonate buffer (pH 9.6). Initial dilutions of sera were prepared in 1% BSA/PBS (pH 7.4), followed by serial dilutions on the plates (100 μL/well, 1% BSA/PBS) for incubation (Dilution range: Pfs25-specific antibody from 1:500 to 1:12,800,000; PfCSP-specific antibody from 1:100 to 1: 5,904,900). Peroxidase-conjugated goat anti-mouse IgG antibody (Seracare, Milford, MA, USA) diluted in 1% BSA/PBS at 1:2000 was used as the secondary antibody (100 μL/well). The plates were developed with ABTS Peroxidase Substrate 1-Component System (Seracare, Milford, MA, USA) at 25 °C for 20 min and then detected at 405 nm wavelength using an ELISA reader (VersaMax, Molecular Devices, San Jose, CA, USA). Antibody endpoint titers were determined as the highest serum dilution that gave an absorbance value greater than the mean plus 3 standard deviations (SD) of the absorbance values of pooled pre-immune sera.

To determine the avidity of antibodies, plates were incubated with NaSCN (0, 0.5, 1, 2, 4 and 8 M) for 15 min after primary antibody incubation, followed by incubation with the secondary antibody and development with ABTS substrate. Binding of antibody to antigen after NaSCN treatment at various concentrations was expressed as a percentage of total binding in the absence of NaSCN. The avidity index was calculated as the NaSCN molar concentration that resulted in 50% dissociation of bound antibody.

IgG isotype analysis was performed using the ELISA as described above using peroxidase-conjugated goat anti-mouse IgG1, IgG2a, IgG2b and IgG3 antibodies (1:5000 dilution, Southern Biotech, Birmingham, AL, USA) as the secondary antibodies. Pooled mouse serum of each DNA-immune group was diluted at the indicated dilutions for IgG isotype analysis. The results are expressed as the ratio of IgG1 to IgG2a.

### 2.5. Standard Membrane Feeding Assays (SMFA)

Mature gametocytes of *P. falciparum* (NF54) were cultured [41] using O+ human erythrocytes at 4% hematocrit in parasite culture medium (RPMI 1640 supplemented with 25 mM Hepes, 10 mM glutamine, 0.074 mM hypoxanthine, and 10% O+ human serum). Gametocyte cultures were initiated at 0.5% parasitemia from low-passage stock and were maintained up to day 18 with daily medium changes. Culture plates were incubated at 37 °C in a microaerophilic environment inside a candle jar. Use of human erythrocytes to support the growth of *P. falciparum* was approved by the Internal Review Board (IRB) of the Johns Hopkins University Bloomberg School of Public Health (#NA 00019050).

Total IgGs were purified using Protein G-Sepharose beads (Invitrogen, Waltham, MA, USA) from pooled sera of immune groups at final bleeds as described previously [27,28]. All purified IgGs were concentrated using Amicon ultra centrifugal filters (MilliporeSigma, Burlington, MA, USA) and adjusted to 8 mg/mL, then stored frozen in PBS. IgGs from Pfs25-EP and combination-EP groups were tested in SMFAs at final concentrations of 2, 1, 0.5 and 0.25 mg/mL. IgGs from the PfCSP-EP group at a final concentration of 1 mg/mL and normal human serum (NHS) were used as negative controls. For SMFA, purified IgGs were serially diluted using NHS and combined with human RBC and *P. falciparum* NF54 gametocytes to 50% hematocrit and 0.2 to 0.3% gametocytemia, respectively. *Anopheles stephensi* mosquitoes (4–5 days old) starved for 6–7 h were fed using water jacketed glass feeders maintained at 37 °C for 15 min. The unfed mosquitoes were removed, and blood-fed mosquitoes were maintained for 8–10 days in the insectary (27 °C, 70–80% RH). Individual mosquito midguts were dissected and stained with 0.5% mercurochrome to determine oocyst number. Mosquito infectivity was measured by comparing oocyst burden as well as the prevalence of infected mosquitoes. Transmission reduction activity (TRA) is defined as the percentage reduction in the number of oocysts, which is calculated using the formula: percentage oocyst reduction = [1 − (mean number of oocysts in test IgG/mean number of oocysts in negative control)] × 100. Transmission blocking activity (TBA) is defined as the percentage reduction in infection prevalence, which is calculated using the formula: percentage prevalence reduction = [1 − (prevalence of infected mosquitoes in test IgG/prevalence of infected mosquitoes in negative control)] × 100.

### 2.6. Sporozoite Challenge and In Vivo Assessment of Liver-Stage Parasite Load and Blood-Stage Parasitemia

To determine protection against sporozoite infection, immunized mice were challenged 2 weeks after the final immunization with a *P. berghei* ANKA transgenic strain (PbPfCSP-GFP/Luc) that expresses PfCSP (3D7 strain) instead of PbCSP, as well as a GFP-luciferase fusion protein [42]. Sporozoites were produced in *A. stephensi* mosquitos that fed on parasite-infected Swiss-Webster mice (Charles River Laboratories, Wilmington, MA, USA). The salivary glands of mosquitoes, 20–22 days post blood feed, were dissected and gently disrupted using pestle and mortar (Kimble, DWK Life Sciences, Millville, NJ, USA) in 2% FBS/HBSS. The released sporozoites were filtered (NITEX Nylon 50 micron mesh, SEFAR, Buffalo, NY, USA) to remove the debris, maintained on ice, counted in a hemocytometer and used within an hour. Mice were challenged intravenously with 2000 fresh PbPfCSP-GFP/Luc sporozoites. Twenty-four hours later, mouse abdominal hair was removed using Nair cream. Forty to forty-two hours after challenge, mice were anesthetized with isoflurane inhalation, injected intraperitoneally with 100 μL of RediJect D-Luciferin (30 mg/mL, PerkinElmer, Boston, MA, USA), and imaged using an in vivo imaging system (IVIS, Lumina III system, PerkinElmer) 10 min after luciferin injection. Parasite liver loads were quantified by analyzing a region of interest (ROI) in the upper abdominal region and determining the radiance flux (photons/sec/cm^2^/sr) expressed by PbPfCSP-GFP/Luc parasites using the manufacturer’s software (Living Image 4.5, PerkinElmer). Vaccine efficacy or parasite load reduction was calculated using the formula: Efficacy or liver-stage parasite reduction = [1 − (Mean flux in test group/Mean flux in the control group)] × 100. Additionally, blood-stage parasitemia in the challenged mice was monitored for up to 2 weeks post challenge by microscopy using thin blood smears prepared daily, fixed with methanol and stained with 10% Giemsa (Sigma-Aldrich, St. Louis, MO, USA). The mice that did not have parasite detected in the blood until 2 weeks post challenge were considered to be fully protected.

### 2.7. Statistical Analysis

Antibody ELISA, mosquito SMFA and blood-stage parasitemia data were analyzed using the Mann–Whitney test. Statistical differences in IVIS data between the control group (PfS25-EP) and all other groups were evaluated by one-way ANOVA and Tukey’s multiple comparison test. The correlation of PfCSP-specific antibody titers and liver-stage parasite loads was determined using linear regression. Differences were considered statistically significant at *p* < 0.05.

## 3. Results

### 3.1. Antibody Responses in the Individual and Combined DNA Plasmid Immunization Groups

Prior to DNA immunization, protein expression of DNA plasmids in mammalian cells was evaluated by transfection of Expi293F cells. Expressed Pfs25 and PfCSP were detected in both supernatants and cell lysates, suggesting the secretion of expressed proteins upon transfection. However, as shown in Appendix A, there were differences between the two proteins, with a relatively higher proportion of Pfs25 being detected in the secreted fractions compared to expressed PfCSP. Western blotting data also showed the presence of reduction-sensitive oligomeric forms of expressed Pfs25.

Sera collected from Balb/c mice in bleed 1, bleed 2 and the final bleed (B1, B2 and FB, Figure 1) were assayed by ELISA to measure both Pfs25-specific and PfCSP-specific antibody responses and monitor antibody kinetics (Figure 2 and Appendix A). After the 3-dose DNA immunization and before the sporozoite challenge (in B2), nine out of ten female Balb/c mice in the PfCSP-EP group exhibited anti-PfCSP antibodies (geometric mean, GM: 46,976), and one mouse had an antibody titer of 900 (Figure 2A). All ten female Balb/c mice in the combination-EP group were positive for anti-PfCSP antibodies (GM: 72,900) (Figure 2A). As expected, no anti-PfCSP response was detected in the female Pfs25-EP group that served as a negative control for anti-PfCSP response (GM: 235). Antibody titers in the combination-EP group were slightly elevated, but not statistically significant when compared with the PfCSP-EP group (*p* = 0.6809), indicating that immunization with DNA combination does not adversely affect the vaccine efficacy of PfCSP DNA (Figure 2A). We observed a similar pattern of PfCSP-specific responses in the male Balb/c mice immunized with PfCSP and Pfs25 alone or in combination (Figure 2B). When comparing antibody response between mouse sexes, the PfCSP-specific antibody titers in male mice were lower than in female mice; however, the differences were not statistically significant (PfCSP-EP, female: 46,976 vs. male: 15,659, *p* = 0.2554; combination-EP, female: 72,900 vs. male: 37,710, *p* = 0.3790). Antibody ELISA data in each bleed (B1, B2 and FB) for all Balb/c groups are shown in Appendix A. Anti-PfCSP antibody titers in FB were found to increase, likely due to heterologous boost by PfCSP proteins expressed on the surface of transgenic PbPfCSP-GFP/Luc used in sporozoite challenge (Appendix A).

For Pfs25-specific antibody response, after 3-dose DNA immunization, all female Balb/c mice in Pfs25-EP and combination-EP groups exhibited anti-Pfs25 antibodies (Figure 2C). The average antibody titer in the Pfs25-EP group was a little higher than that of the combination-EP group, but the difference was not statistically significant (GM, Pfs25-EP: 2,262,742 vs. combination-EP: 1,212,573, *p* = 0.0678) (Figure 2C). We also observed a similar pattern of antibody levels in male Balb/c mice between the Pfs25-EP and combination-EP groups (GM, Pfs25-EP: 1,392,881 vs. combination-EP: 527,803, *p* = 0.3175) (Figure 2D). Similarly to results for PfCSP-specific antibodies, immunization with DNA combination did not compromise the immunogenicity of the Pfs25 DNA vaccine. No anti-Pfs25 response was detected in both female and male Balb/c mice in the PfCSP-EP groups that served as the negative control for anti-Pfs25 response (titers < 1000). Pfs25-specific antibody titers in male mice were lower than those in female mice; however, the differences between male and female mice were not statistically significant (Pfs25-EP, female: 2,262,742 vs. male: 1,392,881, *p* = 0.6154; combination-EP, female: 1,212,573 vs. male: 527,803, *p* = 0.0979) (Figure 2C,D).

Kinetics of anti-Pfs25 antibody responses in various groups are shown in Appendix A. The highest antibody titers were detected in bleeds B1 and B2, which decreased significantly in FB in the Balb/c mice of both Pfs25-EP and combination-EP groups of both sexes. In female Balb/c mice, anti-Pfs25 antibody decreased significantly, by 4.8- and 3.2- fold in the Pfs25-EP and combination-EP groups, respectively (B2 vs. FB; Pfs25-EP: 2,262,742 vs. 466,612, *p* = 0.0007; combination-EP: 1,212,573 vs. 373,213, *p* = 0.005). Antibody titers in the FB of the Pfs25-EP and combination-EP groups were comparable to each other (Appendix A). Similarly, in male Balb/c mice, anti-Pfs25 antibody decreased 3-fold in the Pfs25-EP group; however, the differences were not statistically significant (B2 vs. FB, Pfs25-EP: 1,392,881 vs. 459,479, *p* = 0.1111; combination-EP: 527,803 vs. 459,479, *p* = 0.8810). Finally, anti-Pfs25 antibody titers in the FB were not significantly different between the female and male Balb/c groups (Pfs25-EP: female vs. male, *p* = 0.8127; combination-EP: female vs. male, *p* = 0.7989) (Appendix A). 

Immunogenicity of Pfs25 DNA vaccine was also evaluated in C57BL/6 mice. As shown in Appendix A, the highest anti-Pfs25 antibody titers were detected in bleed 1 (B1, after 2nd DNA dose, GM: 2,111,213) and bleed 2 (B2, after 3rd DNA dose, GM: 2,785,762) (B1 vs. B2, *p* = 0.5238), which were comparable in magnitude to those in female Balb/c mice (Balb/c vs. C57BL/6; B1: 2,910,176 vs. 2,111,213, *p* = 0.3413; B2: 2,262,742 vs. 2,785,762, *p* = 0.7353). For reasons not obvious, a fourth dose of immunization in these C57BL/6 mice resulted in a significant decrease in the anti-Pfs25 antibodies (B2 vs. B3, 2,785,762 vs. 918,959, *p* = 0.0238). Similarly to Balb/c mice, antibody levels decreased further after sporozoite induced blood-stage infection (B3 vs. FB, 918,959 vs. 400,000, *p* = 0.0397). These data demonstrate a similar pattern of antibody responses and kinetics of anti-Pfs25 antibodies in both Balb/c and C57BL/6 mice, ruling out any mouse genetic differences.

To assess the overall strength of antibody–antigen interactions, antibody avidity was measured in the pooled sera of each Balb/c group. No significant difference between single and combination DNA groups was observed in the avidity of anti-Pfs25 and PfCSP antibodies in both female and male mice (Figure 2E). We also extended the analysis for any differences in the isotypes of antibodies among various groups (Figure 2F). Analysis of anti-Pfs25 antibodies in immune sera revealed relatively balanced responses of IgG1 and IgG2a with IgG1/IgG2a ratios varying between 0.6 and 1.3 across groups (between female and male mice and between single and combination groups). However, IgG1/IgG2a ratio values appeared to suggest possible skewing toward higher IgG2a in the Pfs25-EP groups as compared to the combination-EP groups. Isotype analysis of anti-PfCSP antibodies in immune sera revealed predominantly IgG2a-biased responses in all groups, except in the male PfCSP-EP group that exhibited a strong IgG1-biased antibody response (Figure 2F).

### 3.2. Protection against Sporozoite Challenge

The in vivo model of transgenic PbPfCSP-GFP/Luc used in our study represents a valuable tool to characterize protective immunity against PfCSP in preclinical and clinical studies [42,43,44]. Immunized mice were challenged 2 weeks after the final immunization by intravenous injection of 2000 sporozoites, and bioluminescence of the mouse livers was measured to quantify the parasite loads (Figure 3A,B). In the female Balb/c mice, as seen in Figure 3A,C, bioluminescence of the PfCSP-EP group (mean: 25,502 photons/s/cm^2^/sr) and that of the combination-EP group (mean: 14,595 photons/s/cm^2^/sr) were both significantly lower (*p* = 0.0364 and *p* = 0.005), respectively, than that of Pfs25-EP group (negative control group, mean: 70,811 photons/s/cm^2^/sr). The parasite loads were reduced by 64% in the PfCSP-EP group and 79% in the combination-EP groups. PfCSP DNA immunization without EP did not elicit any protective immunity (PfCSP-noEP: 81,058 photons/s/cm^2^/sr, combination-noEP: 70,654 photons/s/cm^2^/sr) when compared with the female Pfs25-EP (70,811 photons/s/cm^2^/sr) or Pfs25-noEP (84,786 photons/s/cm^2^/sr) groups (Figure 3C). One possible explanation may be that anti-PfCSP antibodies were >10-fold lower in the no-EP immune groups (PfCSP-EP vs. PfCSP-noEP: 46,976 vs. 4190; combination-EP vs. combination-noEP: 72,900 vs. 6502) (Appendix A).

We also investigated liver-stage protective immunity in male Balb/c mice to look for any sex differences after DNA immunization (Figure 3B,D). Compared to the Pfs25-EP control group (mean: 41,780 photons/s/cm^2^/sr), the parasite loads were reduced by 62% in the combination-EP group (mean: 16,042 photons/s/cm^2^/sr, *p* = 0.0254) and by 34% in the PfCSP-EP group (mean: 27,508 photons/s/cm^2^/sr, *p* = 0.0573; approaching statistical significance) (Figure 3D). These results demonstrate that PfCSP-specific protective immunity was elicited in both female and male mice, regardless of single or combination DNA vaccines administered by EP. Our studies also ruled out any sex difference in PfCSP-specific antibody responses.

We further evaluated correlations between PfCSP-specific antibodies and liver-stage protective immunity by plotting antibody titers versus parasite loads in the liver (bioluminescence). As shown in Figure 3E,F, there was a significant negative correlation both in female (Figure 3E, R^2^ = 0.5036, *p* < 0.0001) and male mice (Figure 3F, R^2^ = 0.4575, *p* = 0.0056), suggesting that higher PfCSP-specific antibody levels tend to correlate with more protection against sporozoite infection. It also offers an explanation for the higher reduction of parasite loads in the female mice, which revealed relatively higher anti-PfCSP antibodies than in the male mice (female vs. male, PfCSP-EP 64% vs. 34%, combination-EP 79% vs. 62%). 

After the challenge and in vivo imaging, we followed the kinetics of blood-stage parasitemia in mice until 14 days post sporozoite challenge. Mice were considered fully protected if no parasites were detected by day 14. The kinetics of blood-stage parasites are shown using Kaplan–Meier curves and parasitemia curves (Figure 3G,H, and Appendix A). In female mice, both the PfCSP-EP and combination-EP groups exhibited significant delay in the blood-stage parasite growth when compared with the Pfs25-EP control group (PfCSP-EP vs. Pfs25-EP *p* = 0.0117, combination-EP vs. Pfs25-EP *p* = 0.0077, Log-rank test). The kinetics of blood-stage growth between the PfCSP-EP and combination-EP groups were not statistically different (*p* = 0.1044) (Figure 3G). The percentages of parasitemia in the PfCSP-EP and combination-EP groups were also lower than those in the Pfs25-EP control group on each day during infection (Appendix A). One out of 10 mice in the PfCSP-EP group and 2/10 mice in the combination-EP group remained negative for any blood-stage parasitemia, suggesting sterile protection (Figure 3G). PfCSP-specific antibodies in these 3 mice were the highest among all immunized mice (Figure 3E). A mouse in the Pfs25-EP control group with no detectable PfCSP-specific antibodies (Figure 3E) also did not develop a blood-stage infection, possibly suggesting mouse-to-mouse variability of intravenous sporozoite challenge. We also observed significant parasite growth delay in male mice immunized with PfCSP-EP and combination-EP when compared against the Pfs25-EP group (PfCSP-EP vs. Pfs25-EP *p* = 0.0495, combination-EP vs. Pfs25-EP *p* = 0.0143, Log-rank test) (Figure 3H). These results rule out any significant parasite growth differences between PfCSP-EP and combination-EP (*p* = 0.1729) (Figure 3H). Parasitemia percentages in the PfCSP-EP and combination-EP groups were lower than that in the Pfs25-EP group on each day during detection (Appendix A).

### 3.3. Transmission Reducing Activity of Anti-Pfs25 Antibodies

Transmission reducing activity (TRA) of purified IgGs from Balb/c mouse sera in the FB was evaluated by SMFAs in *A. stephensi* mosquitoes (Figure 4). Reduction of oocyst number in mosquitoes was observed in the presence of IgG from the Pfs25-EP group of female Balb/c mice in a dose-response manner (98.4%, 95.4%, 57.3% and 35.8% at 2, 1, 0.5 and 0.25 mg/mL of final IgG concentrations, respectively), with statistically significant transmission reduction demonstrated at 2 and 1 mg/mL when compared to oocyst number in the control group. Moreover, the reduction of infection prevalence (TBA) was also shown to be 72.5% and 48.9% at 2 and 1 mg/mL IgG. Antibodies from the combination-EP group of female Balb/c mice also exhibited significant transmission reduction of 98.9%, 95.8% and 73.6% at 2, 1 and 0.5 mg/mL IgG concentrations, respectively, with corresponding prevalence reduction (TBA) of 61.7%, 47.2% and 14.4%. In contrast, IgG from the PfCSP-EP group that served as an additional immunization control exhibited no significant transmission reduction (27.3% at 1 mg/mL, *p* = 0.7664). (Figure 4A). 

We also tested purified IgGs from male Balb/c mice in SMFAs. Once again, IgGs from the Pfs25-EP and combination-EP groups exhibited transmission reduction in a dose–response manner. IgG of male Pfs25-EP mice resulted in significant transmission reduction (TRA) of 98.4%, 98.5% and 89.5% at 2, 1 and 0.5 mg/mL concentrations, respectively, with corresponding prevalence reduction (TBA) of 74.2%, 75.6% and 59.4%. IgG of male combination-EP mice likewise exhibited significant transmission reduction (TRA) of 99.6%, 98.8% and 97.1%, as well as prevalence reduction (TBA) of 92.7%, 80.9% and 80.1% at 2, 1 and 0.5 mg/mL, respectively. No transmission reduction was observed in IgG from male PfCSP-EP mice (Figure 4B). IgGs from male mice in the Pfs25-EP and combination-EP groups revealed higher TRA and TBA than the female mice, especially at low IgG concentrations (1 and 0.5 mg/mL). This may be either due to higher transmission reducing effectiveness of IgG in the male mice or more likely a result of biological variability of infectivity of culture-derived *P. falciparum* gametocytes used in independent SMFAs testing IgG from male and female mice.

In addition, purified IgG from female C57BL/6 mice immunized with Pfs25 DNA by EP also exhibited significant transmission reduction (TRA) of 99.1% and 98.6% at 2 and 1 mg/mL concentrations, respectively, with corresponding prevalence reduction (TBA) of 84.5% and 83.8% (Appendix A). It is significantly noteworthy that IgGs for all SMFAs were purified in pooled sera from the final bleed, and, in spite of an up to 5-fold drop in antibody titers in Balb/c mice and 7-fold drop in C57BL/6 mice versus the corresponding highest antibody titers (Appendix A), the sera retained potent transmission blocking activity.

Our data clearly demonstrate that anti-Pfs25 antibodies elicited by immunization with Pfs25 DNA alone or in combination with PfCSP DNA have a significant effect on transmission reduction. The similar TRA and TBA between the Pfs25-EP and combination-EP groups, along with similar anti-Pfs25 antibody levels between them, indicate no difference in the functional activity of antibodies induced by single Pfs25 DNA or combination DNA vaccines. Furthermore, the similar effect on transmission reduction shown in both sexes of Balb/c mice ruled out sex as a major biological variable for induction of transmission blocking anti-Pfs25 antibody responses (Figure 4). The similar effect on transmission reduction of anti-Pfs25 antibodies from Balb/c and C57BL/6 mice likewise indicated no difference in immunogenicity and functional activity induced by the Pfs25 DNA vaccine in mice of different genetic backgrounds (Appendix A).

## 4. Discussion

DNA vaccines are considered to be a valuable platform for vaccine development, due to rapid design, ease of production, low cost, high stability, sustainable antigen expression, intrinsic DNA elements to activate innate immunity, and induction of both cellular and humoral immunity [22,45]. When it comes to the development of multiple epitopes or multiple antigens into a single vaccine as well as a combination of multiple vaccines in a single formulation, DNA vaccines have unique advantages that may facilitate the development of malaria vaccines that combine multiple antigens from various lifecycle stages [23]. To date, malaria vaccine candidates that have been tested in humans did not elicit sufficient cellular immunity or antibody responses to achieve complete protection or a transmission blocking effect [8]. Therefore, a combination of multiple, even partially effective, antigens into a vaccine that targets various malaria lifecycle stages may result in higher overall vaccine effectiveness and final outcomes. The immunogenicity and functional activity of Pfs25 and PfCSP DNA vaccines have been demonstrated by us and in other previous studies [24,25,26,27,28,30,31,32,46,47]. In this study, we investigated the immunogenicity and functional activity of these two leading candidates using a DNA vaccine platform in mice, individually or in a combination. In vivo EP was used for DNA immunization, because EP has been shown to markedly improve the immunogenicity of DNA vaccines without any known side effects. Other nucleic acid-based immunization platforms, especially the mRNA-LNP based vaccines, have recently shown their flexibility, rapidity and effectiveness against COVID-19 [48]. We have also initiated studies to evaluate the immunogenicity and functional activity of the combination of PfCSP and Pfs25 based on mRNA-LNP vaccines in mice (unpublished data).

Immunization with combined DNA vaccines conferred comparable anti-PfCSP antibody responses in mice of both sexes when compared to those with single PfCSP DNA immunization. Protective immunity induced either by single PfCSP DNA or by combined DNAs reduced the parasite loads in the livers of immune mice challenged with transgenic PbPfCSP-GFP/Luc, which was demonstrated using in vivo imaging and subsequent monitoring of blood-stage parasitemia. We found a positive correlation between anti-PfCSP antibody levels and protective immunity. However, our studies did not exclude the contribution of cellular immunity that might also be induced to confer protection in the PfCSP DNA immunization, as reported previously [23,30,46,47,49]. Particularly noteworthy was the observation that a combination of PfCSP DNA with Pfs25 DNA did not compromise the immunogenicity and protective immunity induced by PfCSP DNA. We did not observe any protection against parasite challenge in the mice immunized with PfCSP DNA vaccine (single and combined DNAs) without EP, although previous studies reported that PfCSP DNA immunization by intramuscular injection elicited specific antibody and cellular immune responses in animals [31,37]. This may be due to weak antibody responses in these no-EP immune mice in our study, which could not confer protection in the in vivo challenge model. Our studies further corroborate prior reports that in vivo EP markedly enhances the immunogenicity and functional activity of the DNA-based malaria vaccine [26,50].

When we analyzed anti-Pfs25 antibody responses after three DNA doses, immunization with combined DNA vaccines also elicited similar levels of anti-Pfs25 antibody in mice of both sexes, when compared to immunization with Pfs25 DNA alone. Although anti-Pfs25 antibodies waned largely in mice that experienced sporozoite-induced blood-stage infection, significant transmission reduction was still achieved in SMFAs using purified IgGs from mouse sera obtained at the final bleed. Comparable effects on TRA and TBA between single Pfs25 DNA and combined DNAs implies that the combination of DNA vaccines does not negatively affect the antibody response and functional activity induced by Pfs25 DNA. As we reported before [24], much lower anti-Pfs25 antibody responses were observed in the mice immunized without EP by single and combined DNAs (Appendix A). We did not test the sera of these mice in SMFAs because of the expected low TRA or TBA. 

No significant differences in avidity of anti-Pfs25 or anti-PfCSP antibodies were observed between single and combined DNA immunizations, indicating that DNA combination did not affect the binding strength of specific antibodies to their respective antigens. Antibody isotype analysis revealed some IgG1-biased and IgG2-biased skewing in immunization of combined DNAs. The combined DNAs promoted the IgG1 skewed responses in anti-Pfs25 antibodies and the IgG2a skewed responses in anti-PfCSP antibodies. However, we did not observe any related changes in the functional activity of anti-Pfs25 antibodies in SMFAs and anti-PfCSP antibodies in the immune mice after in vivo challenge.

Our studies also indicated that female mice produced higher specific antibodies than the male mice, regardless of whether immunized by single or combined DNA vaccines. We observed 2.3-fold higher anti-Pfs25 antibodies induced by combined DNAs in the female mice when compared to male mice, although the difference did not reach statistical significance (*p* = 0.0979). Anti-Pfs25 antibodies induced by single Pfs25 DNA in the female mice were also 1.6-fold higher than those in the male mice (*p* = 0.6154). These antibody titer differences did not seem to impact the actual functional activity in SMFAs. Similarly, anti-PfCSP antibodies in the female mice were 3-fold and 1.9-fold higher than those in the male mice when immunized with single PfCSP DNA and combined DNAs, respectively. 

Taken together, the findings of our study demonstrate that a combination of Pfs25 and PfCSP DNAs into a vaccine with delivery by EP into mice induced potent immune responses against both Pfs25 and PfCSP. The specific antibodies elicited by combined DNAs were similarly effective in protecting against infection and reducing mosquito transmission when compared to Pfs25 DNA or PfCSP DNA alone. Taking the advantages of DNA vaccines into consideration, this may represent a powerful and cost-effective platform for developing a combined DNA vaccine with multi-stage malaria antigens for further evaluation in nonhuman primates and humans. In particular, a combined DNA vaccine targeting different life stages of *Plasmodium* may optimize overall effectiveness when compared to a single antigen vaccine and may prove to be a valuable tool for targeting malaria transmission.

## Figures and Tables

**Figure 1 vaccines-10-01134-f001:**
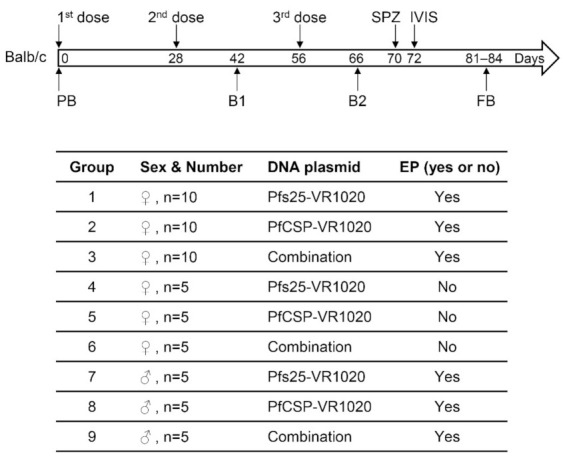
Schematic representation of immunization, sera collection, sporozoite challenge and IVIS. Nine groups of Balb/c mice, both female and male, were immunized by single (Pfs25 or PfCSP) or combined (Pfs25 plus PfCSP) DNA vaccines with or without in vivo EP at the indicated time points, followed by sporozoite challenge and in vivo imaging (IVIS). Various bleeds analyzed are also indicated. PB, pre-immune bleed; B1, bleed 1; B2, bleed 2; FB, final bleed.

**Figure 2 vaccines-10-01134-f002:**
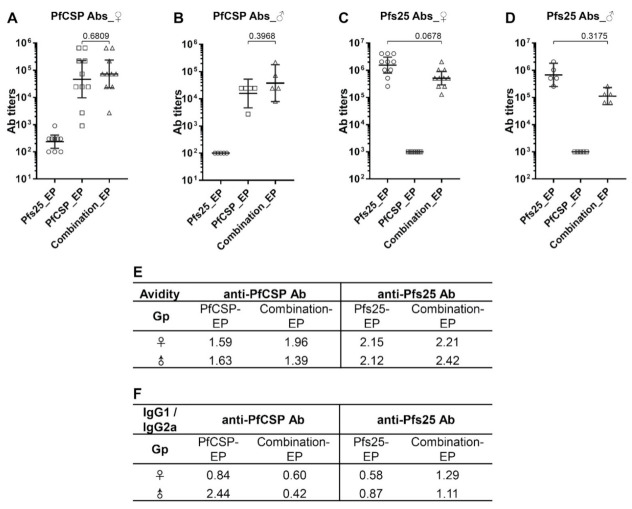
Evaluation of antibody titers, avidity and isotype by ELISA. Sera of Balb/c mice after 3-dose DNA immunization were assayed for Pfs25-specific and PfCSP-specific antibodies using recombinant proteins of Pfs25 and PfCSP. Antibody titers are reported as the geometric mean (GM) with 95% confidence intervals (CI). (**A**,**B**) PfCSP-specific antibody responses in female (**A**) and male (**B**) Balb/c mice. (**C**,**D**) Pfs25-specific antibody responses in female (**C**) and male (**D**) Balb/c mice. Statistical analyses were performed using the Mann–Whitney test, and *p* values are indicated. (**E**) Antibody avidity was assayed in the pooled mouse sera of each group, and the numbers indicate NaSCN molar concentration required for 50% dissociation of bound antibodies. (**F**) Antibody isotypes were tested in pooled mouse sera of each group and compared as a ratio of IgG1 to IgG2a.

**Figure 3 vaccines-10-01134-f003:**
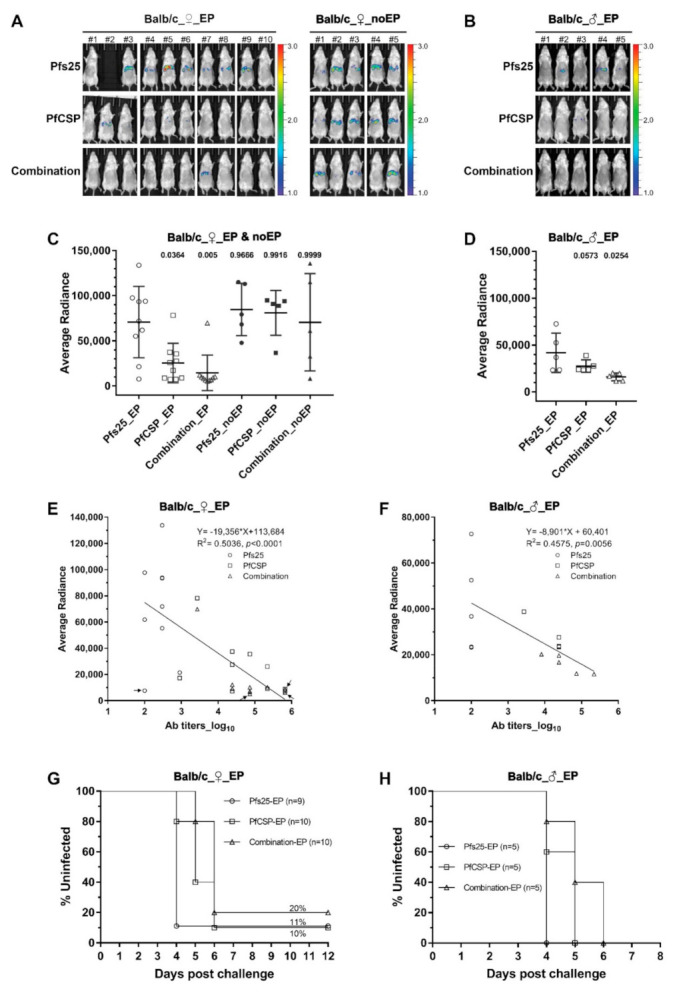
Protection against sporozoite challenge. Balb/c mice immunized with single and combined DNA vaccines were challenged by intravenous injection of 2000 sporozoites (PbPfCSP-GFP/Luc) per mouse 2 weeks after the third immunization. At 40–42 h after challenge, mice were injected intraperitoneally with D-Luciferin and imaged using IVIS. (**A**,**B**) Images of in vivo bioluminescence in mouse livers. Rainbow scales are expressed in radiance flux (×10^5^ p/sec/cm^2^/sr). (**A**) Female mice in groups 1–6, immunized with or without EP. (**B**) Male mice in groups 7–9, immunized with EP. (**C**,**D**) Quantification of total flux from female mice (shown in (**A**)) and male mice (shown in (**B**)), respectively. Bars indicate the arithmetic mean and SD, and each symbol represents an individual mouse. Statistical differences between the control group (Pfs25-EP) and all other groups were evaluated by one-way ANOVA and Tukey’s multiple comparison test. *p* values are indicated on the top. (**E**,**F**) Correlations between PfCSP-specific antibody ELISA titers and liver-stage parasite loads (bioluminescence) for female mice in groups 1–3 (**E**) and male mice in groups 7–9 (**F**), respectively. Each symbol represents an individual mouse. Four mice with no detectable parasite 14 days after challenge are indicated by arrows in E, indicating full protection from challenge. Correlation was determined using a linear regression. Equations, R^2^ values and *p* values are shown. (**G**,**H**) Kaplan–Meier survival curves reporting percentage of parasite-free mice after challenge. (**G**) Female mice in groups 1–3. (**H**) Male mice in groups 7–9. Mice were monitored for blood-stage parasitemia by Giemsa-stained thin blood smears each day post challenge. Sterile protection was defined as mice without detectable parasitemia in the blood during the 14-day follow-up.

**Figure 4 vaccines-10-01134-f004:**
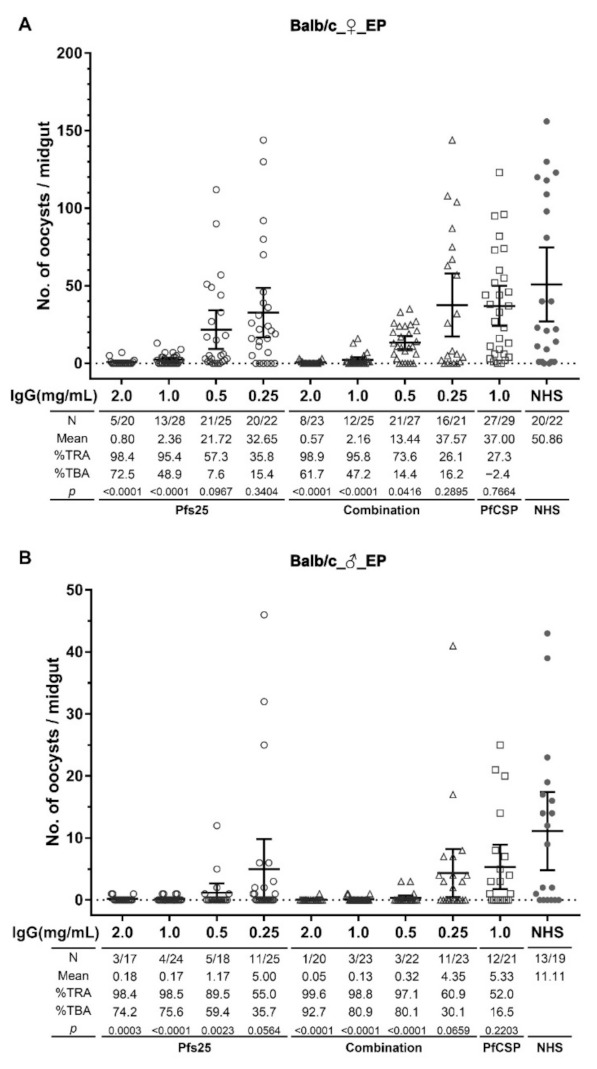
Transmission reducing activity of Pfs25-specific antibodies in Balb/c mice immunized with single and combined DNA vaccines. Total IgGs purified from pooled immune sera of each Balb/c group were assayed in SMFAs at final IgG concentrations of 2, 1, 0.5 and 0.25 mg/mL (*x*-axis). Mosquitoes fed without purified IgG (NHS) were used in SMFAs as the negative control. IgG from the PfCSP group was also tested at a fixed 1 mg/mL concentration and served as an independent negative control. Each symbol represents the number of oocysts in an individual mosquito, and the bars indicate the arithmetic mean and the 95% confidence interval (95% CI). (**A**) SMFA results with IgGs from female Balb/c mice in groups 1–3, immunized by single or combined DNA vaccines with EP. (**B**) SMFA results with IgGs from male Balb/c mice in groups 7–9, immunized by single or combined DNA vaccines with EP. Analysis details are summarized in the tables beneath each panel. N represents the numbers of mosquitoes with midgut oocysts/total numbers of mosquitoes dissected. Average numbers of oocysts in each group are shown as mean values. % TRA and %TBA were calculated as described in the materials and methods section. Statistical analyses were performed using the Mann–Whitney test, and *p* values are shown.

## Data Availability

Not applicable.

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
