# Peer review of "Effective Functional Immunogenicity of a DNA Vaccine Combination Delivered via In Vivo Electroporation Targeting Malaria Infection and Transmission"

_vaccines, 2022, doi:10.3390/vaccines10071134_

Round 1

Reviewer 1 Report

Work by Ciao et al. describes the immunogenicity of combination of two malaria vaccine candidates using a DNA vaccination approach. Authors tested a pre-erythrocytic (PfCSP) and a transmission blocking (Pfs25) candidate tested alone and in combination, and protection as defined by reduced infection following a sporozoite challenge as well as transmission reducing/blocking activities was assessed. They show that the combination of PfCSP and Pfs25 DNAs in mice does not compromise immunogenicity, infection protection and transmission reduction compared to each vaccine tested individually. Overall, these findings support that combination of malaria vaccines with different targets is maybe an optimal strategy to evaluate. My comments below.

1-      Data reported here seem to suggest that no synergetic effect of the combination was detected, contrary to work by Sherrard-Smith et al., https://doi.org/10.7554/eLife.35213 albeit monoclonal antibodies were used as vaccine surrogate.  The authors should discuss this difference.

2-      The authors should also discuss their DNA vaccine approach in a context of novel nucleic acid based vaccine approach that, now day, are proven to be efficacious and more implementable than DNA vaccines that may require EP. I am also intrigued by the IgG isotype bias in male vs female and feel the authors should discuss this as well.

3-      It is not clear how the authors ruled out sex as a biological variable for induction of transmission blocking based on the difference in TRA/TBA at the lower concentrations of IgG. Please provide any statistical analysis performed in support of this claim.

Line 83: Pfs25 instead?

Line 210: Please indicate the dilution range

Line 371: host genetic?

Author Response

Reviewer 1

Comment 1: Data reported here seem to suggest that no synergetic effect of the combination was detected, contrary to work by Sherrard-Smith et al., https://doi.org/10.7554/eLife.35213 albeit monoclonal antibodies were used as vaccine surrogate.  The authors should discuss this difference.

Response 1: In the Sherrard-Smith’s study, synergy of two monoclonal antibodies (mAb) in reducing parasites was tested in a multi-generational population assay using a murine malaria model and passive transfer of mAbs. The overall goals of that study and our study are conceptually similar, i.e, to evaluate either “functional synergy” or “additive” potential of immune responses targeting two different stages during malaria infection and transmission. However, the approaches used are very different. In the Sherrard-Smith study they used titrated amounts of passively transferred monoclonal antibodies (one against Pfs25 and the other one against PbCSP) using a murine model to assess any functional synergy. Our studies employed active immunization with Pfs25 and PfCSP immunogens. Our studies demonstrate that when used for immunization in a combination, the overall immunogenicity and functional efficacy of individual immunogens was not compromised. Unfortunately, there is no convenient animal model system for P. falciparum (human malaria parasite) to evaluate using a multi-generational population approach. However, our studies clearly establish that a combination of DNA vaccines induces immune responses that target infection by sporozoite and transmission by gametocytes and similar active immunization will be expected to decrease overall transmission and infection prevalence at the population level. A direct evidence for either additive or synergistic outcome over multiple malaria transmission cycles will be provided by actual vaccine trials in endemic human population.

Comment 2: The authors should also discuss their DNA vaccine approach in a context of novel nucleic acid based vaccine approach that, now day, are proven to be efficacious and more implementable than DNA vaccines that may require EP. I am also intrigued d by the IgG isotype bias in male vs female and feel the authors should discuss this as well.

Response 2: We have briefly mentioned other nucleic acid-based vaccination approaches, especially the mRNA-LNP vaccines. There are ongoing unpublished studies from our own lab which are revealing the value of using mRNA-LNP vaccine platform for malaria. Our studies using DNA vaccines delivered via in vivo electroporation established uncompromised immunogenicity of combination vaccines, and it is to be expected that similar combination of mRNA-LNP or DNA vaccines administered using other approaches such as gene gun, lipid nano-formulations and needle-free system may be employed in future to achieve the targeted functional goal.

In the isotypes analysis of anti-Pfs25 antibodies, we observed that IgG1/IgG2a ratios showed skewing toward higher IgG2a in Pfs25-EP groups (both male and female mice) as compared to the combination-EP groups. However, that did not seem to affect the functional transmission reducing activity (TRA) between the Pfs25-EP and combination-EP groups. In the isotypes analysis of anti-PfCSP antibodies, we observed strong IgG1 bias in the male PfCSP-EP group in contrast to IgG2a bias in all other groups, however, the difference was not associated with any protection difference against sporozoite infection. These are interesting observations and will require dedicated sex-based investigation in future.

Comment 3: It is not clear how the authors ruled out sex as a biological variable for induction of transmission blocking based on the difference in TRA/TBA at the lower concentrations of IgG. Please provide any statistical analysis performed in support of this claim.

Response 3: We showed that anti-Pfs25 antibody ELISA titers in the final bleed (FB) used for purification of IgG in SMFA were not significantly different between female and male mice in Pfs25-EP group and combination-EP group. These purified IgG from male and female mice were tested at multiple concentrations (2.0 to 0.25 mg/ml) separately in two independent SMFAs using different batches of cultured P. falciparum gametocytes. The actual oocyst counts between SMFAs for IgG from male and female cannot be compared because of intrinsic biological variability between different batches of parasite cultures used in independent mosquito feeds. However, we observed similar potent TRA at two high IgG concentrations (2 and 1 mg/ml) in both female and male mice in Pfs25-EP group and combination-EP group, and no significant TRA difference at the lowest IgG concentrations (0.25 mg/ml) in all these groups. Only at the IgG concentration of 0.5 mg/ml, Pfs25-EP group and combination-EP group of male mice showed slightly higher TRA/TBA as compared to IgG from female mice. Taken together, both male and female mice responded to immunization (ELISA titers) and exhibited similar comparable TRA/TBA in SMFAs, thus supporting our conclusion that mouse sex is not a major biological variable for induction of transmission blocking anti-Pfs25 antibody responses.

Responses to other minor comments:

Line 83: Pfs25 instead? : Modified in manuscript.

Line 210: Please indicate the dilution range: Modified in manuscript.

Line 371: host genetic? : Modified in manuscript.

Reviewer 2 Report

I have reviewed the manuscript titled " 

Effective functional immunogenicity of DNA vaccine 2

combinations delivered via in vivo electroporation targeting 3 malaria infection and transmission by Cao Y et al. Manuscript describes the protective potential of two Plasmodium vaccine candidate antigens alone or in combination; CSP, a sporozoite surface antigen and a mosquito stage antigen, Pfs25 by DNA immunization through intramuscular in vivo immunization. I recommend the publication of this manuscript with minor revisions. Although data presented is fine, I am not sure that how feesible will be to try this technique (EP)  in the field setting. Moreover authors should have compared these results with Prime-boost immunization as they already have the mammalian expressed proteins.  Minor suggestions

1. Authors should provide a schematics of two prteins and their regions expressed in Supplementary figure 1.

2. Like-wise authors should have provided schematics of the genes used in immunization along with vector maps in supplementary figures

3. Although authors have compared the protective potential and parasitemia in only antigen(s) immunized mice vs in combination, I do not see much differences in single antigen vs combination. authors should discus this in discussion section.

Author Response

Reviewer 2

Comment 1: ……………..Although data presented is fine, I am not sure that how feasible will be to try this technique (EP) in the field setting.

Response 1: We used the in vivo electroporation (EP) for DNA immunization, because in our previous published studies we have shown EP to significantly improve immunogenicity of DNA vaccines. Using DNA delivery with in vivo EP allowed us to evaluate the feasibility of immunization using combination vaccines. It should be noted the in vivo EP delivery of DNA vaccines is subject of a number of clinical trials. However, the feasibility of EP for large scale field trials such as those that will be required for vaccination against malaria will have to be re-assessed. The key conceptual evidence supporting developing combination vaccines provided by our studies (DNA delivery via EP) is expected to pave the way for further evaluation of DNA combination vaccine using other delivery methods such as, gene gun, lipid-based carriers and needle-free injection system. Additionally, the use of mRNA-LNP which does not depend upon delivery methods such as EP may offer potent alternative approaches.

Comment 2: Moreover, authors should have compared these results with Prime-boost immunization as they already have the mammalian expressed proteins.

Response 2: The two DNA vaccine plasmids, PfCSP-VR1020 and Pfs25-VR1020, were used for transfection of the mammalian cells to simply verify protein expression and secretion of two vaccine antigens under the mammalian conditions. Our studies support immunization with DNA vaccines alone to be quite effective and hence we did not plan to employ heterologous prime-boost approaches. We don’t disagree that a prime-boost approach may further improve immunogenicity outcomes, and may be a subject for future investigation.

Comment 3: Although authors have compared the protective potential and parasitemia in only antigen(s) immunized mice vs in combination, I do not see much differences in single antigen vs combination. Authors should discuss this in discussion section.

Response 3: Our studies reveal that the combination of PfCSP and Pfs25 DNA vaccines induced potent immune responses against both antigens, and specific antibodies were similarly effective in protecting against infection and reducing mosquito transmission when compared to single DNA vaccine alone. Protection against infection induced by PfCSP DNA singly or in combination was tested using the in vivo sporozoite challenge, whereas transmission reducing activity of anti-Pfs25 antibodies was tested in SMFA. Whether immunization with a combination will provide “synergistic” or “additive” outcomes has been discussed in response to a comment (Comment 1) from Reviewer 1 (please see above).

Minor (reviewer 2) suggestions

  1. Authors should provide a schematics of two proteins and their regions expressed in Supplementary figure 1.
  2. Like-wise authors should have provided schematics of the genes used in immunization along with vector maps in supplementary figures.

Response: Two DNA vaccine plasmids, PfCSP-VR1020 and Pfs25-VR1020, were used in the previous studies. The schematics of DNA vaccines representing the genes (PfCSP and Pfs25) and DNA vector (VR1020) were published in the papers from our and other research groups, which are cited in our manuscript. The coding sequence regions used for cloning downstream of TPA signal sequence in the VR-1020 plasmids and for expression in E. coli as 6xHis-tagged proteins are clearly stated in the materials and methods section.